# Explorative Sonophotocatalytic Study of C-H Arylation Reaction of Pyrazoles Utilizing a Novel Sonophotoreactor for Green and Sustainable Organic Synthesis

Tamer S. Saleh [1,2,*] , Abdullah S. Al-Bogami [1], Katabathini Narasimharao [3] , Ziya A. Khan [1] , Iban Amenabar [4] and Mohamed Mokhtar [3,*]

1   Department of Chemistry, College of Science, University of Jeddah, Jeddah 21959, Saudi Arabia
2   Green Chemistry Department, National Research Centre, Dokki, Giza 12622, Egypt
3   Chemistry Department, Faculty of Science, King Abdulaziz University, P.O. Box 80203, Jeddah 21589, Saudi Arabia
4   nanoGUNE BRTA and Department of Electricity and Electronics, UPV/EHU, 20018 Donostia-San Sebastián, Spain
*   Correspondence: tssayed@uj.edu.sa (T.S.S.); mmoustafa@kau.edu.sa (M.M.)

**Abstract:** The development of a mild, general, and green method for the C-H arylation of pyrazoles with relatively unreactive aryl halides is an ongoing challenge in organic synthesis. We describe herein a novel sonophotoreactor based on an ultrasonic cleaning bath and blue LED light (visible light) that induce copper-catalyzed monoarylation for pharmacologically relevant pyrazoles. The hybrid effect of ultrasonic irradiation and blue LED is discussed to interpret the observed synergistic action. A broad array of pyrazoles coupled with iodobenzene avoids expensive palladium metal or salts, and certain designed substrates were attained. Only comparatively inexpensive copper(I)iodide and 1,10-phenanthroline were used all together as the catalyst. The presented technique is a greener way to create C-H arylation of pyrazoles. It significantly reduces the amount of energy needed.

**Keywords:** ultrasound irradiation; sonophotoreactor; blue LED; C-H arylation; pyrazoles; copper iodide; 1,10-phenanthroline

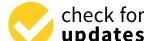

## 1. Introduction

The exploration of green chemistry methodologies is a challenge that is framed by the "twelve principles of green chemistry" [1]. These principles identify catalysis as one of the fundamental pillars for implementing the concept of green chemistry. Catalysis is very important for both organic and medicinal chemists. It introduces a plethora of green chemistry benefits during organic synthesis [2] such as lower energy requirements, improved selectivity, and reduced use of processing [3,4]. Catalytic operations can be performed through one of the various green chemistry techniques such as ultrasonic irradiation, leading to spectacular results [5]. Green chemistry and sonochemistry are multidisciplinary. Eco-friendly processes based on ultrasound have been reported in many areas such as sonocatalysis, organic chemistry, preparation of materials, polymer chemistry, biomass conversion, extraction, electrochemistry, enzymatic catalysis, and environmental remediation [6–9]. According to previous reports, there are three available types of ultrasonic reactors in the market of scientific devices: cleaning baths, direct immersion ultrasonic horns (DIUH), and ultrasonic probe cup horns [6]. Noticeably, most organic chemists carry out their organic reactions through a sonoreactor such as a cleaning bath. Cleaning baths with unique characteristics in terms of the power, vessel size, number of transducers, and distribution of the cavitational activity were used [10–14]. The second reactor design, direct immersion ultrasonic horns (DIUHs), is avoided in organic synthesis. The reluctance to utilize DIUH in organic synthesis is due to the tremendous energy released, which is

incompatible with the covalent bonds found in organic molecules; therefore, it will be modified to use just 25% of its power if used [15]. Furthermore, when it is used in organic synthesis, the titanium tip of the horn undergoes erosion, raising a cautionary limitation for this type [16]. The third kind of reactor is the ultrasonic probe cup horn, which is mostly used to remove organic contaminants from water (for wastewater treatment) [17,18]. Additionally, a large-diameter ultrasonic probe cup horn is sometimes employed in chemical synthesis [19]. Nonetheless, it has several drawbacks, including the fact that it is only suitable for laboratory-scale operations and that the high power wasted inside the reaction may be changed at a particular frequency. In terms of cavitational activity and application, the diameter of the probe and the height of the liquid in the reactor are the determining variables [20]. As a result, the ultrasonic cleaning bath's unique specification makes it the optimal reactor design for organic synthesis. Noteworthily, there are certain drawbacks to this kind of design. It is possible to address these issues, such as differences in acoustic intensity between cleaning baths made by different manufacturers, but this may be overcome via calibration since the calibration of equipment is a prerequisite for laboratory certification under ISO/IEC 17025 [21]. Consequently, the issue of reproducibility between baths may be managed and regulated. Other important factors should be taken into account when optimizing reaction conditions, for example, the location, form of the reaction flask and the reaction vessel's internal temperature [22–25].

The only green chemistry method that is considered more ignored is photochemistry, despite the fact that light is non-toxic and traceless [26]. There are many reasons for this, the most significant of which, in our view, is that photochemical reactions are uncommon in the industry. As a result, the environmental component has been overlooked in detail, as the majority of instances have included only small-scale exploratory investigations [27]. Additionally, as is usually the case with green photochemical processes, the lamp and solvent selection are critical [27–29]. Different economical light sources, such as LEDs, may have become beneficial during the past decade [30]. This kind of light has been evaluated in a large variety of chemical processes, with substantial yield and/or selectivity increases reported [31–34]. Additionally, LEDs overcome some of the constraints associated with costly and sometimes dangerous light sources employed in photochemical reactions [34,35]. Blue LED light may be used to activate non-toxic catalysts, allowing for the performance of processes that previously required harsh or toxic chemical reagents. There are persistent and dedicated efforts being made to eradicate the industry's disdain for photochemistry methods by addressing their drawbacks and limits, particularly economic ones [36]. However, the majority of efforts do not attempt to create a method for scaling up organic processes. Rather than that, the majority of these efforts focus on overcoming the limitations of costly and dangerous sources of light, which they accomplish to a degree. Combining several techniques into a single instrument, in our view, may benefit both economic and environmental concerns. In a recent report, the synergistic impact of two sources of energy (ultrasound and light) on the catalysis process was discussed. This is considered major research since it presents a sonophotoreactor based on a cup horn sonicator for the oxidation process [37]. This new research serves as a testbench for comprehending the unusual hybrid effects of ultrasound and photochemistry. The reported new finding motivated us to develop an efficient design and fabricate a novel sonophotochemical reactor by using an ultrasonic cleaning bath and a blue LED for organic synthesis reactions to realize our dream of providing an economic method that may be scaled up without implying a negative connotation for organic synthesis in this regard. We provide an efficient design and complete experimental setup for a new sonophotoreactor based on an ultrasonic cleaning bath and a blue LED in this article. The C-H arylation of many pyrazoles was used as a model reaction to allow us to fully establish the novel sonophotoreactor experimentally. As a catalyst, this suggested model example makes use of copper(I)iodide in the presence of phenanthroline. The purpose of these experimental designs is to maximize the usage of this new sonophotocatalytic reactor and/or the reaction selectivity, which aids in planning the process's scaling up.

## 2. Results and Discussion

### 2.1. The Developed Cleaning Bath Sonophotoreactor (CBSPR) and Its Characterization

We previously succeeded in determining the best operational parameters for utilizing ultrasonic cleaning baths in different organic reactions using various heterogeneous and homogeneous catalysts [38–43]. These operating parameters relating to the choice of cleaning baths for organic reactions, preserving a specific temperature during its working, and the shape and position of the reaction vessels critically affect the organic reactions.

Then, the characteristics and positioning of the light source and heat-transfer properties are critical for the outcome of the studied chemical transformation. A schematic illustration of our designed, complete, and functional cleaning bath sonophotoreactor referred to as a CBSPR is represented in Figure 1. According to the Beer–Lambert law, light transmittance decreases exponentially with the distance from the light source [44].

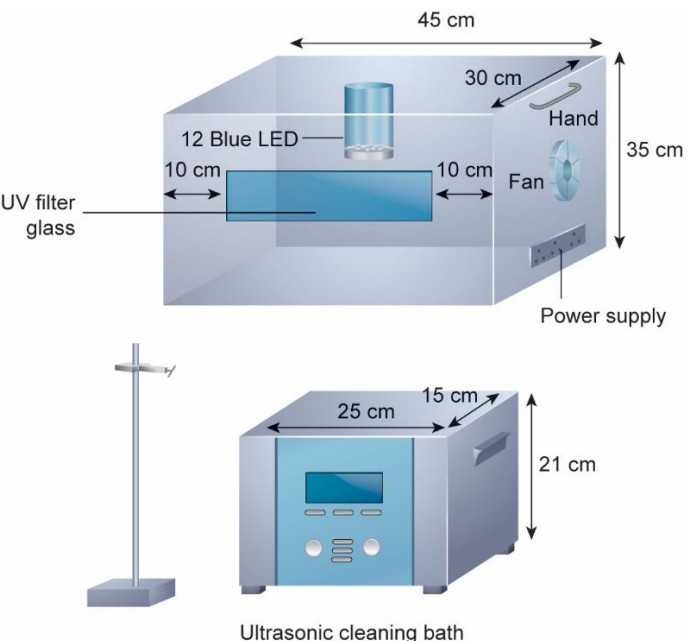

**Figure 1.** A schematic illustration of our designed cleaning bath sonophotoreactor (CBSPR).

Therefore, we consider that the light intensity reduced considerably from the flask wall into the bulk of the reaction. Accordingly, a demo of the experimental setup was made, considering the light intensity reduces significantly from the flask walls to the middle of the reaction mixture, resulting in slow reactions and nonhomogeneous irradiation. Therefore, we selected a significant organic reaction such as the C-H arylation of pyrazole as a model reaction to fine-tune the experimental setup. The reaction is represented in Scheme 1.

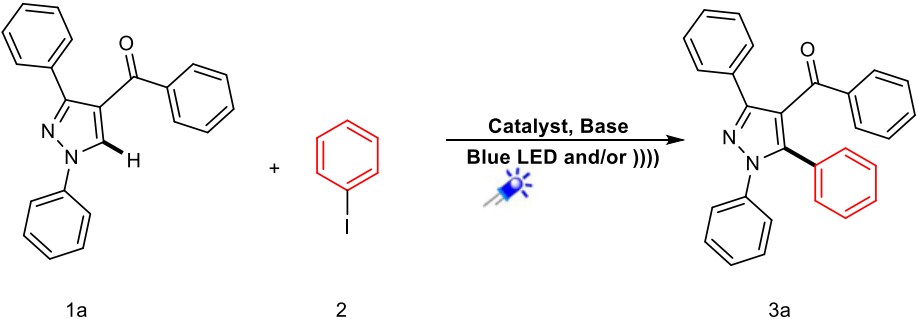

**Scheme 1.** Optimization of the reaction conditions for the C-H arylation of pyrazole. First, we carried out this reaction under various conditions, including catalyst, ultrasound irradiation, and/or blue LED light; optimization of the reaction conditions was conducted at several stages, including the main stage, which is the catalytic activity study.

Catalytic Activity Study

As Budiman et al. demonstrated, the combination of CuI and 1,10-phenanthroline, CsF, in DMF solvent at 130 °C is efficient for the Suzuki–Miyaura cross-coupling of fluorinated aryl boronate esters with aryl iodide [45]. For the above model reaction depicted in Scheme 1, Table 1 represents different experimental trials to get the best conditions. We initially tried to employ modified conditions to the C-H arylation of pyrazole **1a** with phenyl iodide (**2**), utilizing ultrasound irradiation at 80 °C in presence of 10 mol% CuI and obtained only a 39% isolated yield of the C-H arylated pyrazole **3a** after 3.5 h (Table 1, entry 6). This model reaction was carried out on the same scale utilizing the blue LED light only at 80 °C, we separated a yield of **3a** as 31% after 9 h (Table 1, entry 7). The reaction performed on the designed sonophotoreactor led to a better yield in the shortest reaction time in which we obtained a 48% yield in 2 h of arylated pyrazole **3a** (Table 1, entry 8).

**Table 1.** Synthesis of compound **3a** using different catalysts under ultrasonic irradiation, blue LED light, and sonophotoreactor conditions.

| Entry | Conditions | Catalyst | Solvent | Base (Equiv.) | Temp. | Time of Reaction | % Yield |
|---|---|---|---|---|---|---|---|
| 1 | Ultrasound (US) | No catalyst | DMF | CsF (2) | 80 °C | 4 h | 0 |
| 2 | Blue LED | No catalyst | DMF | CsF (2) | 80 °C | 10 h | 0 |
| 3 | US | 10 mol% 1,10-phenanthroline | DMF | CsF (2) | 80 °C | 8 h | 0 |
| 4 | Blue LED | 10 mol% 1,10-phenanthroline | DMF | CsF (2) | 80 °C | 8 h | 0 |
| 5 | US and blue LED | 10 mol% 1,10-phenanthroline | DMF | CsF (2) | 80 °C | 8 h | 0 |
| 6 | US | 10 mol% CuI | DMF | CsF (2) | 80 °C | 3.5 h | 39 |
| 7 | Blue LED | 10 mol% CuI | DMF | CsF (2) | 80 °C | 9 h | 31 |
| 8 | US and blue LED | 10 mol% CuI | DMF | CsF (2) | 80 °C | 2.5 h | 48 |
| 9 | US | 10 mol% CuI, 10 mol% phenanthroline | DMF | CsF (2) | 80 °C | 2.5 h | 63 |
| 10 | Blue LED | 10 mol% CuI, 10 mol% phenanthroline | DMF | CsF (2) | 80 °C | 7 h | 55 |
| 11 | US and blue LED | 10 mol% CuI, 10 mol% phenanthroline | DMF | CsF (2) | 80 °C | 2 h | 81 |
| 12 | US and blue LED | 10 mol% CuI, 10 mol% phenanthroline | DMF | $K_2CO_3$(2) | 80 °C | 1.5 h | 82 |
| 13 | US and blue LED | 10 mol% CuI, 10 mol% phenanthroline | DMF | $K_2CO_3$ (1) | 80 °C | 1.5 h | 80 |
| 14 | US and blue LED | 10 mol% CuI, 10 mol% phenanthroline | DMF | $K_2CO_3$ (3) | 80 °C | 1.5 h | 82 |
| 15 | US and blue LED | 10 mol% CuI, 10 mol% phenanthroline | EtOH | $K_2CO_3$ (2) | 80 °C | 1.5 h | 80 |
| 16 | US and blue LED | 10 mol% CuI, 10 mol% phenanthroline | DCM | $K_2CO_3$ (2) | 80 °C | 1.5 h | 65 |
| 17 | US and blue LED | 10 mol% CuI, 10 mol% phenanthroline | 1,4-Dioxan | $K_2CO_3$ (2) | 80 °C | 1.5 h | 68 |
| 18 | US and blue LED | 10 mol% CuI, 10 mol% phenanthroline | DMF | Pot. acetate (2) | 80 °C | 2.5 h | 75 |
| 19 | US and blue LED | 10 mol% CuI, 10 mol% phenanthroline | DMF | Pot. phosphate (2) | 80 °C | 3 h | 70 |

It is evident from the results represented in Table 1 that CuI with phenanthroline as a catalyst has an essential role in this reaction. No reaction was observed in the absence of it when utilizing ultrasound alone (Table 1, entry 1) or blue LED (Table 1, entry 2). The same result was observed when we used 10 mol% of phenanthroline by itself under ultrasonic irradiation (Table 1, entry 3), blue LED (Table 1, entry 4), or both ultrasound and blue LED at the same time (our designed reactor) (Table 1, entry 5). However, when we used both 10 mol% of CuI and 10 mol% phenanthroline as catalyst under ultrasonic irradiation, a 63% yield was obtained in 2.5 h (Table 1, entry 9). When both 10 mol% of CuI and 10 mol% phenanthroline were used in our designed reactor, we attained a 79% yield of the pure arylated pyrazole 3a in 2 h only utilizing the two equivalents of CsF as base Table 1, entry 11). As it turned out, the best ligand for our needs was phenanthroline. While a variety of bases were tested (Table 1, entries 12, 18,19), two equivalents of potassium carbonate was found to be the most effective in the model reaction (Table 1, entry 12). An interesting side note is that, in testing these model reactions, we found that DMF performed better than ethanol, dichloromethane (DCM), and 1,4-dioxane as a reaction solvent (Table 1, entries 12, 15–17). Ultrasound and blue LED radiation have a synergistic effect, as shown by the results presented here (the designed sonophotoreactor). Using the sonophotoreactor, the time was slashed from 2.5 or 7 h under ultrasound irradiation or blue LED, respectively, to 1.5 h utilizing the designed sonophotoreactor. Testing different catalyst loadings was used to fine-tune the reaction settings using the CuI/phenanthroline catalyst at different mol%, and thin-layer chromatography (TLC) under the designed sonophotoreactor was used to monitor the reaction's progress. The results attained from the catalytic test reaction are mentioned in Table 2.

**Table 2.** Optimization of reaction conditions for the synthesis of **3a** utilizing CuI and phenanthroline catalyst.

| Entry | Catalyst mol% | Yield | Time | Blue LED Distance (cm) |
|:---:|:---:|:---:|:---:|:---:|
| 1 | 5 mol% CuI, 5 mol% phenanthroline | 78% | 2 h | 2 |
| 2 | 10 mol% CuI, 10 mol% phenanthroline | 82% | 1.5 h | 2 |
| 3 | 15 mol% CuI, 15 mol% phenanthroline | 82% | 1.5 h | 2 |
| 4 | 10 mol% CuI, 10 mol% phenanthroline | 68% | 3 h | 1 |

Initially, for optimizing the mass of catalyst, diverse mol% of CuI and phenanthroline catalyst were tested. As a result, in 1.5 h process runs using 10% catalyst, approximately the optimum product yield was observed (Table 2, entry 2). Moving to examine the impact of the distance of the light source from the reaction flask on this catalytic system, two experiments were conducted with the 10 mol% CuI, 10 mol% phenanthroline catalyst at 1 and 2 cm (entries 2, 4). We found that performing this sonophotochemical reaction in a short-necked round-bottomed 50 mL flask and the presence of a blue LED source at a distance of 2 cm from the neck of the flask allowed a higher and more homogeneous photon flux, resulting in shorter reaction times and consequently high selectivity (no side-product formation) due to avoiding over-irradiation.

The structure of the formed arylated pyrazole **3a** was confirmed on the basis of elemental analysis and spectral data; especially, $^1$H NMR showed the disappearance of the CH pyrazole proton singlet signal at δ 7.9 ppm and the appearance of new aromatic multiplets (cf. Supporting Information S2). We studied the beneficial effects of our reactor design on the synthesis of compound **3a** by performing the reaction at reflux conditions under the same scale as described above, but poor yield was obtained with a longer reaction time (41% in 16 h), and a higher temperature was used.

Noteworthily, the reproducibility of the designed sonophotoreactor was tested by extending the reaction protocol for several derivatives using copper chitosan catalyst under the above optimized conditions (Scheme 2).

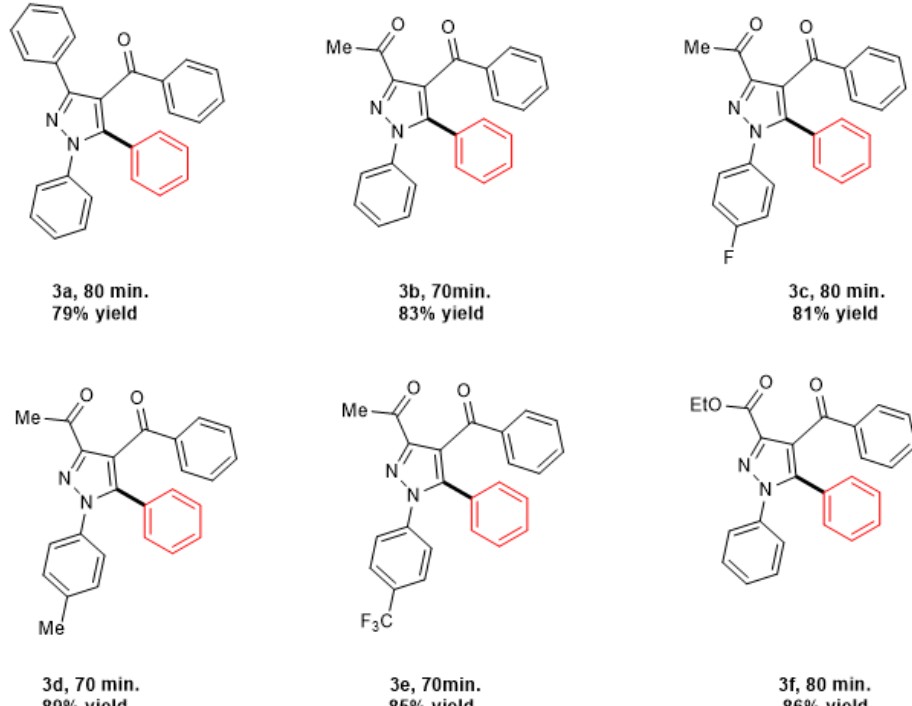

**Scheme 2.** Sonophotocatalytic C-H arylation of pyrazoles.

The products **3a-h** were obtained at acceptable yields (79–89%) (Figure 2), which shows the beneficial effects of the sonophotoreactor on this reaction. The obtained spectroscopic data of the reaction products **3a-h** and the proper elemental investigation reinforced the pyrazole structure formed in each case for **3a-h** (cf. Supporting Information S2). There are numerous studies showing that cavitation acoustic waves, which are responsible for the formation of bubbles in liquid, are responsible for increasing the rate and selectivity of sonochemical reactions [46–49]. In addition, the cavitation intensity increases with increasing frequency and solvent type. For the sonochemical reaction to be successful, the solvent used must be carefully chosen. The vast majority of procedures are carried out in a wet environment. As an alternative, we utilized dimethyl formamide (DMF) in our reaction. DMF has a high vapor pressure at 70–80 °C [50], which is ideal for cavitation. The low viscosity of DMF makes cavitation easier [51,52]. As a result of these findings, it can be concluded that DMF is a better solvent than the others.

3a, 80 min.
79% yield

3b, 70min.
83% yield

3c, 80 min.
81% yield

3d, 70 min.
89% yield

3e, 70min.
85% yield

3f, 80 min.
86% yield

**Figure 2.** *Cont.*

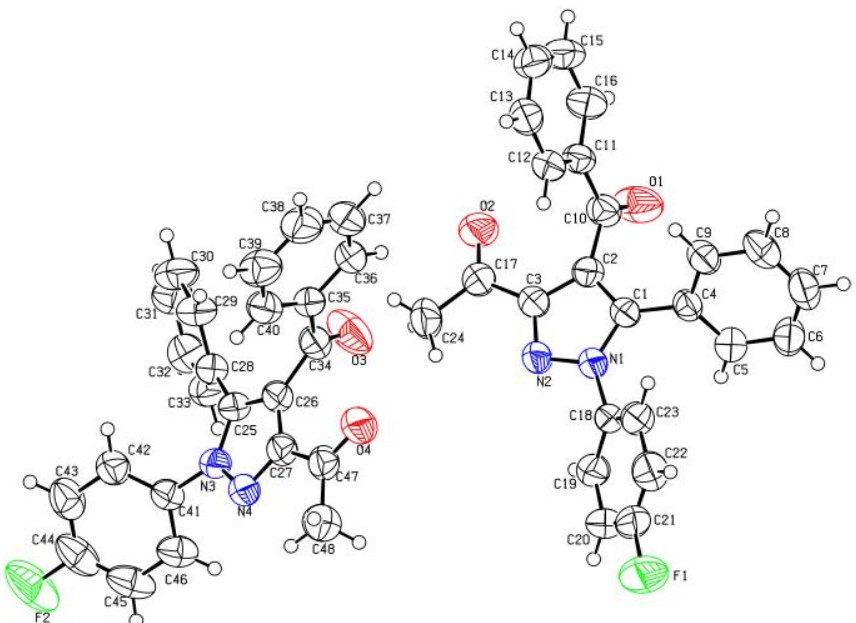

**Figure 2.** Structure of pyrazole synthesized via C-H arylation utilizing the designed sonophotoreactor.

The proposed structure of CH arylated pyrazoles **3c** and **3e** was further unequivocally confirmed by X-ray crystallography (Figures 3 and 4), single-crystal X-ray diffraction of compounds **3c** and **3e** adds sharp evidence for the proposed structure [53,54].

**Figure 3.** Single-crystal X-ray of compound **3c**.

A tentative mechanism was proposed for the synergistic effect observed in the sonophotoreactor based on the well-established mechanism for the effect of ultrasound in the presence of homogeneous catalysts such as complexes [55,56]. In this case, an alkali metal base deprotonated a relatively acidic sp$^2$ C-H bond before transmetalation and coupling with an aryl halide were performed on it [55,56]. The known mechanism for the effect of visible light on catalytic reactions involving complexes (photocatalytic reactions) [57] was also taken into consideration, which involved single electron transfer (SET). Herein, the reaction proceeded by initial deprotonation of a relatively acidic sp$^2$ C-H bond by an alkali metal base (K$_2$CO$_3$) forming an intermediate A, which was further enhanced by ultrasonic irradiation (step I, Scheme 3). Then, the formation of the copper metal complex (intermediate B) via transmetalation was ultrasonically enhanced. The complex (B) was excited by blue-light absorption to give the intermediate B*, which was followed by single electron transfer (SET) to phenyl iodide, by effect of blue LED, generating copper II complex (C) and phenyl radical. Consequently, the phenyl radical, in the presence of the proper radical acceptor, formed the functionalized pyrazole and regenerated the catalytical competent copper(I) complex (Scheme 3). Notably, the observed synergistic effect is attributed mostly to the change in the number of cavitating bubbles due to an increasing degree of coalescence. The results also suggested that sonophotocatalytic reactions occur at the bubble–liquid

interface. In the center of the bubble, the concentration of free radicals "produced by the effect of the blue LED" is at its highest point, and it gradually declines in a Gaussian way as one approaches the interphase owing to radical recombination [58]. In the interphase, the phenyl radical reacts rapidly with the intermediate C, resulting in a buildup of the sonoproduct concentration.

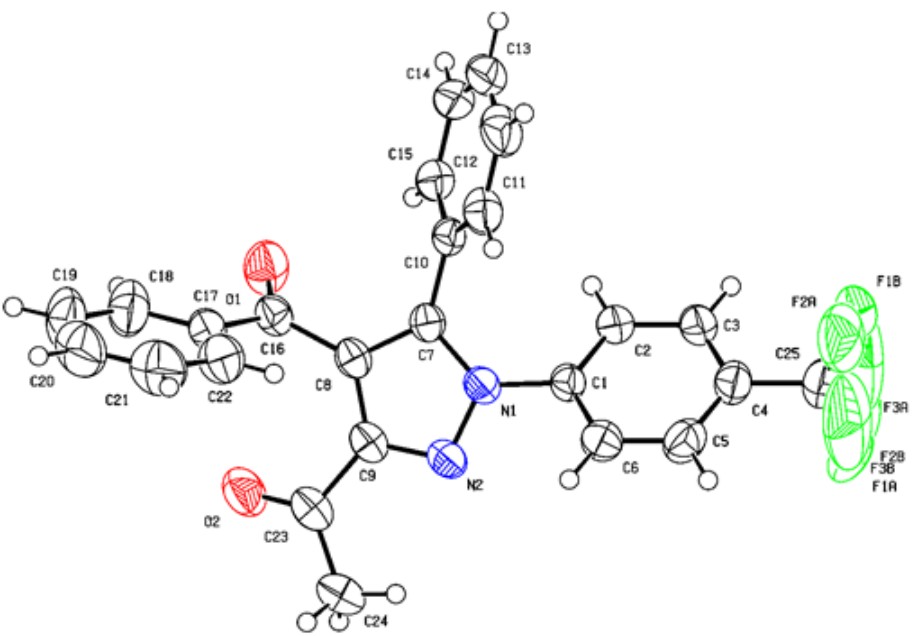

**Figure 4.** Single-crystal X-ray of compound **3e**.

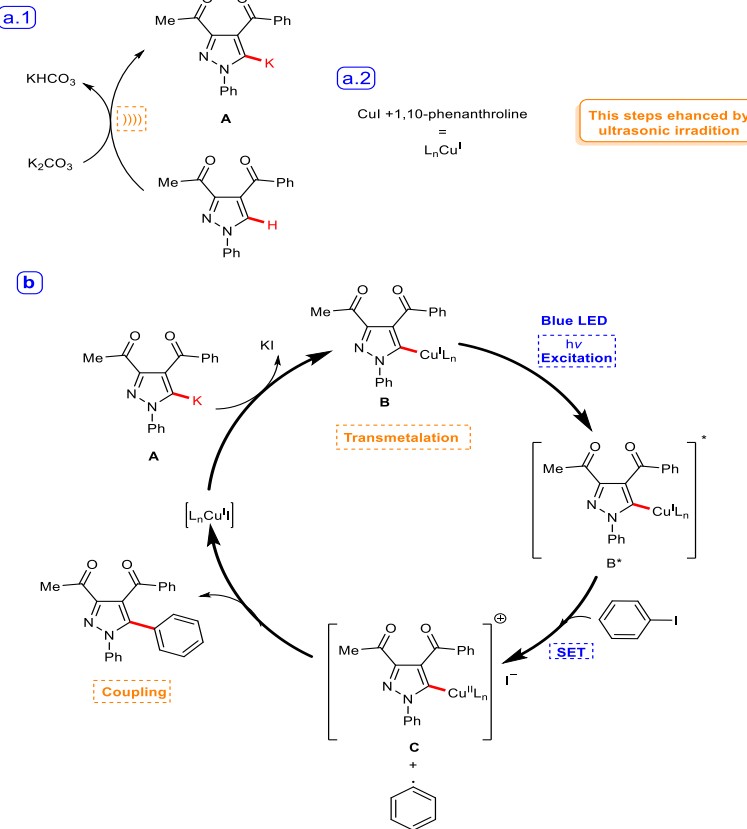

**Scheme 3.** A tentative mechanism for sonophotocatalytic CH arylation of pyrazole using copper complex as a catalyst in the sonophotocatalytic process.

## 3. Experimental

### 3.1. General

All organic solvents were purchased from commercial sources and used as received unless otherwise stated. All other chemicals were purchased from Merck, Aldrich, or Acros and used without further purification. Thin-layer chromatography (TLC) was performed on precoated Merck 60 GF254 silica gel plates with a fluorescent indicator, and detection was achieved by means of UV light at 254 and 360 nm. The melting points were measured on a Stuart melting point apparatus and are uncorrected. IR spectra were recorded on a Smart iTR, which is an ultra-high-performance, versatile attenuated total reflectance (ATR) sampling accessory on the Nicolet iS10 FT-IR spectrometer(Thermo Fisher Scientific, Waltham, MA, USA). The NMR spectra were recorded on a Bruker Avance III 400 (Bruker, Billerica, MA, USA) (9.4 T, 400.13 MHz for $^1$H, 100.62 MHz for $^{13}$C analysis) spectrometer with a 5 mm BBFO probe at 298 K. Chemical shifts (δ in ppm) are given relative to internal solvent, DMSO-d$_6$ 2.50 for $^1$H and 39.50 for $^{13}$C. Mass spectra for synthesized derivatives were recorded using a Thermo ISQ Single-Quadrupole GC-MS instrument(Waltham, MA, USA). Elemental analysis was carried out using a Euro Vector EA3000 Series C, H, N, S analyzer (Pavia, Italy). Pyrazole 1a-i derivatives were prepared according to methods reported in the literature [59–61]. Sonication was performed using an Elma Sonicator P30H instrument (Singen, Germany) with an ultrasound frequency of 37 kHz and power of 320 W (max). The temperature of the bath was raised from 25 to 80 °C after 30 min of operation. All the reactions were carried out at 80 °C, which was maintained by adding or removing the water in ultrasonic bath (the temperature inside the reaction vessel was 76–79 °C). The dimension of the sonophotoreactors is depicted in Figure 1, the blue LED lamp used in the designed reactor was an EvoluChem™ light source designed specifically for photocatalytic chemistry applications. The wavelength was 425 nm and was designed with a beam angle of 25° to focus the light toward the reaction sample.

Compounds **3c** and **3e** were recrystallized for single crystal X-ray diffraction studies. This sample was mounted on an Agilent SuperNova (Dual source) Agilent Technologies Diffractometer (USA) for each sample, equipped with microfocus Cu/Mo Kα radiation for data collection. The data collection was accomplished using CrysAlisPro software [62] at 296 K under the Mo Kα radiation. The structure solution was performed using SHELXS–2013 method [63] and refined by full–matrix least–squares methods on F2 using SHELXL–2013 method [63], in-built with WinGX [64]. All non–hydrogen atoms were refined anisotropically by full–matrix least squares methods [62].

### 3.2. General Procedures for Synthesized Arylated Pyrazole 3a-h Derivatives

#### 3.2.1. Sonicated Reactions

A mixture of pyrazole **1a-h** (1 mmol), iodobenzene 2 (1 mmol), and K$_2$CO$_3$ (2 mmol) in DMF (10 mL) containing 10 mol% CuI and 10 mol% 1,10-phenanthroline was sonicated using an Elma sonicator P30H instrument. All the reactions were carried out at 80 °C, which was maintained by adding or removing the water in ultrasonic bath (the temperature inside the reaction vessel was 76–79 °C). The sonochemical reactions were continued for a determined time until the starting materials were no longer detectable by TLC. Water (5 mL) was added, and the mixture was extracted with ethyl acetate (3 × 5 mL). The combined organic layers were dried over sodium sulfate and concentrated under reduced pressure followed by silica gel column chromatography purification to afford the compounds **3a-h**.

#### 3.2.2. Blue LED Reactions

The reactions under blue LED light were performed on the same scale and conditions as described for sonicated reactions. All the reactions were carried out at 80 °C, and the reactions were continued for a determined time (until the starting materials were no longer detectable by TLC). After completion of the reaction, the products obtained were purified as described previously for the sonicated reaction.

### 3.2.3. Sonophotochemical Reaction

The reactions under sonophotochemical conditions were performed on the same scale and conditions as described for both sonicated and blue LED reactions. All the reactions were carried out at 80 °C, and the reactions were continued for a determined time (until the starting materials were no longer detectable by TLC). After completion of the reaction, the products obtained were purified as described previously for the sonicated reaction.

### 3.2.4. Reflux Condition

The reactions under reflux condition were performed on the same scale and conditions as described for both sonicated, blue LED, and sonophotochemical reactions for the synthesis of **3a**. The reactions carried out under reflux were continued for a determined time (until the starting materials were no longer detectable by TLC). After completion of the reaction, the products obtained were purified as described previously for the sonicated reaction, which gave only 41% yield in 16 h.

The physical and spectroscopic data of all synthesized compounds were given in supporting information.

### 4. Conclusions

We demonstrated the design and construction of a lab sonophotoreactor based on a cleaning bath sonicator and LED light sources that can be easily scaled up. Using pyrazole C-H arylation as a model reaction, the most important parameters were optimized. Synergistic action on simultaneous irradiation by two separate sources of power, light, and ultrasound at the same time has been shown to have an important effect on the selectivity of the C-H arylation of pyrazoles nucleus. This may be due to acoustic cavitation's unique physical and mechanical effects, such as the formation of hot spots and microjets, which play an important role in the controllable mass transfer. Since the technique depicted can be used in the synthesis of small drug molecules, our research efforts are focused on this direction. Scaled-up reactors could also benefit from this new system's "green and sustainable" approach.

**Supplementary Materials:** The following supporting information can be downloaded https://www.mdpi.com/article/10.3390/catal12080868/s1. S1.1: The synthesized compounds with their physical data; S1.: Single-crystal X-ray determination; S2: An illustration of our design; S3: Copies of NMR data; S4: checkCIF (basic structural check) running; S5: checkCIF (basic structural check) running.

**Author Contributions:** T.S.S. Suppose the idea and research design, T.S.S. and A.S.A.-B. carried out all the experiments, K.N. and Z.A.K. analyzed the data, T.S.S. and I.A. were mainly responsible for writing the original draft of the paper; M.M. and T.S.S. carried out writing—review and editing. All authors have read and agreed to the published version of the manuscript.

**Funding:** This research was funded by the Deputyship for Research and Innovation, Ministry of Education in Saudi Arabia, grant number 1066.

**Data Availability Statement:** Not applicable.

**Acknowledgments:** The authors extend their appreciation to the Deputyship for Research and Innovation, Ministry of Education in Saudi Arabia for funding this research work through the project number 1066.

**Conflicts of Interest:** The authors claim that they do not have any known conflicting financial interests or personal affiliations that may seem to have impacted the work presented in this study.

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
