# Peer review of "Explorative Sonophotocatalytic Study of C-H Arylation Reaction of Pyrazoles Utilizing a Novel Sonophotoreactor for Green and Sustainable Organic Synthesis"

_catalysts, doi:10.3390/catal12080868_

Round 1
Reviewer 1 Report
The authors deals with the development of a sono/photochemical apparatus to accomplish a C-H arylation reaction on pyrazoles.
In general the manuscript is well written and clear through all the section. The strategy is adequately presented as well as executed. In addition the apparatus developed, in combination with the use of heterogeneous catalyst shows a wise and logical adoption of a ''green'' way of think novel process.
In this regard are my only few minor concerns:
1) unfortunately, the use of DMF is damaging the overall grennes of the reported procedure. Can the author made some screening of green solvent, this would be for sure enhance the quality of the work
2) Still regarding the solvent usage, can the authors show a table with a molarity screening, the use of 30 mL for 1 mmol is to much also for photochemistry
After addressing this minor issues the manuscript can be accepted
Author Response
Reviewer: 1
- unfortunately, the use of DMF is damaging the overall grennes of the reported procedure. Can the author made some screening of green solvent, this would be for sure enhance the quality of the work
Response
Thank you for the reviewer comment, We try already some other green solvents such water, Ethanol but the results not good, but we will extend our work in future to involve the ionic liquids as green solvents.
- Still regarding the solvent usage, can the authors show a table with a molarity screening, the use of 30 mL for 1 mmol is to much also for photochemistry
Response
Thank you for the reviewer comment, We try doing experiment again with less amount of solvent and we get same results when we use 10 mL. (We modify this in experimental part).

Reviewer 2 Report
In this manuscript entitled "Explorative Sonophotocatalytic Study of C-H Arylation Reaction of Pyrazoles Utilizing a Novel Sonophotoreactor for Green Sustainable Organic Synthesis", the authors presented a copper-catalyzed arylation of pyrazoles with photo- and sono-activation. They built a sono-photoreactor to realize their reactions and showed its efficiency on 8 substrates. However, the reaction described is already well-described with palladium as catalyst but also with copper (See JACS 2008, 130, 15185). In addition, the use of ultrasound and blue LED doesn't bring a significant improvement (table 1: entries 9, 10 and 11). Otherwise I was wondering if there is a mistake on scheme 2 because "Copper chitosan" is indicated as catalyst and all the optimisations were done with CuI. Concerning the proposed mechanism (scheme 3), the generation of a phenyl radical is questionable. A mechanism with oxidative addition and reductive elimination (Chin. J. Chem. 2020, 38, 1299) would be more suitable. Finally, in the supporting information some corrections should be done:
. compound 3b: signal at 7,77 should be for 2H.
. compound 3c: signal at 7,78 should be for 2H. Some coupling C/F should be detected in C NMR.
. compound 3d: signal at 7,80 should be for 2H.
. compound 3e: a quadruplet should be seen for the CF3.
. compound 3g: Some coupling C/F should be detected in C NMR.
For all these reasons, I would suggest to refuse this article.
Author Response
Reviewer: 2
In this manuscript entitled "Explorative Sonophotocatalytic Study of C-H Arylation Reaction of Pyrazoles Utilizing a Novel Sonophotoreactor for Green Sustainable Organic Synthesis", the authors presented a copper-catalyzed arylation of pyrazoles with photo- and sono-activation. They built a sono-photoreactor to realize their reactions and showed its efficiency on 8 substrates. However, the reaction described is already well-described with palladium as catalyst but also with copper (See JACS 2008, 130, 15185).
Response
Thank you for the reviewer comment, We used the reaction described in our manuscript to characterize our reactor, our reactor give satisfied and excellent yield and shortest reaction time than mentioned in literature.
In addition, the use of ultrasound and blue LED doesn't bring a significant improvement (table 1: entries 9, 10 and 11).
Response
Thank you for the reviewer comment, there were obvious improvement in using both US and blue LED together than Using individual on reaction time and % yield. Using Blue LED only (7h), US only (2.5h), Both (1.5-2h) and % yield increased from 55% into 82%.
Otherwise I was wondering if there is a mistake on scheme 2 because "Copper chitosan" is indicated as catalyst and all the optimisations were done with CuI.
Response
Thank you for the reviewer comment, we correct this error in the revised version.
Concerning the proposed mechanism (scheme 3), the generation of a phenyl radical is questionable. A mechanism with oxidative addition and reductive elimination (Chin. J. Chem. 2020, 38, 1299) would be more suitable.
Response
Many Thanks for the reviewer comment, really this add to our manuscript, I accept your esttemed suggestion but formation of copper (III) due to double electron transfer as mentioned in Chin. J. Chem. 2020, 38, 1299 and other reference such Pure Appl. Chem. 2014; 86(3): 345–360 is less common in presence of photocatalysis than formation of copper (II). But single electron transfers more common in photochemistry, we mentioned in this manuscript a tentative mechanism, but now we do such reaction with heterogenous catalyst contains Cu(I) and study the oxidation states of during reaction to prove the mechanism with XPS and different state of art techniques such sSNOM.
Finally, in the supporting information some corrections should be done:
compound 3b: signal at 7,77 should be for 2H.
Done.
compound 3c: signal at 7,78 should be for 2H. Some coupling C/F should be detected in C NMR.
Done.
compound 3d: signal at 7,80 should be for 2H.
Done
compound 3e: a quadruplet should be seen for the CF3.
Done but we see only 1JCF as sometimes fluorinated carbons are difficult to find in 13C spectra with low signal-to-noise ratios because the signal is spread over multiple lines and can be buried in the noise.
compound 3g: Some coupling C/F should be detected in C NMR.
Done

Reviewer 3 Report
The authors developed a lab sonophotoreactor for the Cu-I catalyzed C-H arylation pyrazoles with aryl iodides as reactants. The authors have optimized several parameters in this study and found the 10% CuI, 10% 1,10-phenanthroline as the efficient catalytic system under ultrasonic irradiation and blue LED. This study has significant novelty for the publication in the catalyst after addressing the following comments.
Comments:
1. What is the reaction yield with 10 mol% CuI and 10 mol% phenanthroline under conventional heating (DMF reflux or 100 oC).
2. Performing the reaction under radical scavengers like TEMPO would provide key insights into the reaction mechanism.
3. Does this method limit to only trisubstituted pyrazines? Does arylation occur at both positions if the phenyl(1-phenyl-1H-pyrazol-4-yl)methanone is subjected to reaction conditions? How about the selectivity of this substrate?4. The authors should include examples of other heterocycles, such as indoles/triazoles in the transformation.
5. The authors claim the purpose of this study is to maximize the usage of a sonophotocatalytic reactor for process scale-up. However, they have not performed large-scale reactions in this setup? They should include an example of a gram scale reaction.
6. The complete reaction conditions should be included in scheme 2.
7. As the reaction is sensitive to the temperature and position of the vessel, the image of the reaction setup should be provided. This could be highly beneficial to the scientific community.
Other minor text corrections:
Rephrase the paragraph.
a) Page 2 line 65,
On the other hand, although photochemistry may be called a green chemistry 65 method due to the fact that light is non-toxic and traceless reagent, it is considered the 66 more ignored green chemistry technique.
b) Correct the sentence 2.5 h min
Line 161
the time is slashed from 2.5h min. or 7h under ultrasound irradiation or blue LED respectively into 1.5h utilizing the designed sonophotoreactor.
Author Response
Reviewer: 3
- What is the reaction yield with 10 mol% CuI and 10 mol% phenanthroline under conventional heating (DMF reflux or 100 oC)?
Response
Thank you for the reviewer comment, we do this experiment under reflux condition and the obtained results as the following for synthesis of 3a, Time 16 h and yield only 41%. We add this results in Result and discussion part, also, Experimental part.
- Performing the reaction under radical scavengers like TEMPO would provide key insights into the reaction mechanism.
Response
Thank you for the reviewer comment, we mention in our manuscript a tentative mechanism based on literature, and we will prove it in our future work using your esteemed idea via utilizing radical scavenger and now we do such reaction with heterogenous catalyst contains Cu(I) and study the oxidation states of during reaction to prove the mechanism with XPS and different state of art techniques such sSNOM.
- Does this method limit to only trisubstituted pyrazines? Does arylation occur at both positions if the phenyl(1-phenyl-1H-pyrazol-4-yl)methanone is subjected to reaction conditions? How about the selectivity of this substrate? & 4.The authors should include examples of other heterocycles, such as indoles/triazoles in the transformation.
Response
Thank you for the reviewer comment, for his important note, we do the reaction mainly to facilitate the characterization of our mentioned reactor, in the studied substrates we only get one product as mentioned and support it via Spectroscopic data and Single crystal X-ray, under our optimized conditions we do not notice any trace amount of other product such arylation of aromatic rings so high selectivity obtained, but we do not try other substrate such phenyl(1-phenyl-1H-pyrazol-4-yl)methanone or other heterocycles, we will put it in our consideration in the future work.
- The authors claim the purpose of this study is to maximize the usage of a sonophotocatalytic reactor for process scale-up. However, they have not performed large-scale reactions in this setup? They should include an example of a gram scale reaction.
Response
Thank you for the reviewer comment, for his important note, we can do in our described reactor large scale experiment till 1.5 gram due to size of round bottom flask used. time in large scale experiment increase which consumed to heat more volume of solvent. A larger scale is available if we used a cleaning bath with larger volume that used here.
- The complete reaction conditions should be included in scheme 2.
Response
Thank you for the reviewer comment, We add it.
- As the reaction is sensitive to the temperature and position of the vessel, the image of the reaction setup should be provided. This could be highly beneficial to the scientific community.
Response
Thank you for the reviewer comment, we already put image of reaction setup in the supporting information part.
Other minor text corrections:
Rephrase the paragraph.
- a) Page 2 line 65,
On the other hand, although photochemistry may be called a green chemistry 65 method due to the fact that light is non-toxic and traceless reagent, it is considered the 66 more ignored green chemistry technique.
Done
- b) Correct the sentence 2.5 h min
Line 161
the time is slashed from 2.5h min. or 7h under ultrasound irradiation or blue LED respectively into 1.5h utilizing the designed sonophotoreactor.
Done

Round 2
Reviewer 3 Report
The manuscript can be accepted in its current form.